# Two Years after *Coxiella burnetii* Detection: Pathogen Shedding and Phase-Specific Antibody Response in Three Dairy Goat Herds

**DOI:** 10.3390/ani13193048

**Published:** 2023-09-28

**Authors:** Christa Trachsel, Gaby Hirsbrunner, T. Louise Herms, Martin Runge, Frederik Kiene, Martin Ganter, Patrik Zanolari, Benjamin U. Bauer

**Affiliations:** 1Clinic for Ruminants, Department of Clinical Veterinary Science, Vetsuisse Faculty, University of Bern, 3012 Bern, Switzerland; christa.trachsel@unibe.ch (C.T.); gabriela.hirsbrunner@unibe.ch (G.H.); 2Lower Saxony State Office for Consumer Protection and Food Safety (LAVES), Food and Veterinary Institute Braunschweig/Hannover, Eintrachtweg 17, 30173 Hannover, Germany; louise.herms@laves.niedersachsen.de (T.L.H.); martin.runge@laves.niedersachsen.de (M.R.); 3Clinic for Swine and Small Ruminants, Forensic Medicine and Ambulatory Service, University of Veterinary Medicine Hannover, Foundation, Bischofsholer Damm 15, 30173 Hannover, Germany; frederik.kiene@tiho-hannover.de (F.K.); martin.ganter@tiho-hannover.de (M.G.); benjamin.bauer@tiho-hannover.de (B.U.B.)

**Keywords:** goat, Q fever, bulk tank milk, dust swab, milking parlor, phase-specific serology, zoonosis

## Abstract

**Simple Summary:**

The bacterium *Coxiella (C.) burnetii* causes Q fever in humans and animals, with ruminants acting as reservoirs and shedding the pathogen during abortion or birth. Inhalation of contaminated aerosols is the main route of transmission to humans in which *C. burnetii* can cause a persistent focalized infection with severe consequences. Goats have been identified as a source of human Q fever. This study aimed to describe the infection dynamics 2 years after initial detection of *C. burnetii* in three goat herds by analyzing vaginal swabs, bulk tank milk, and dust samples from a barn and milking parlor. Antibody responses were measured in sera using phase-specific ELISAs. The results varied among the herds. In one herd, the pathogen was no longer detectable, but some animals had seroconverted. In the other two herds, *C. burnetii* was shed to varying degrees and elevated antibody levels were present, indicating ongoing or past infection. The milking parlor showed the highest degree of contamination, highlighting the risk during milking activities. In conclusion, the risk of *C. burnetii* shedding in dairy goat herds persists 2 years after the first detection, and dust swabs from a milking parlor can serve as an easy sampling tool.

**Abstract:**

The infection dynamics of *Coxiella (C.) burnetii* were investigated in three dairy goat herds (A, B, and C) 2 years after the first pathogen detection. A total of 28 and 29 goats from herds A and B, and 35 goats from herd C, were examined. Sera were analyzed on three sampling dates using phase-specific serology. Pathogen shedding was assessed using post-partum vaginal swabs and monthly bulk tank milk (BTM) samples. Dust samples from a barn and milking parlor were also collected monthly. These samples were analyzed with PCR (target IS*1111*). In herd A, individual animals tested seropositive, while vaginal swabs, BTM, and most dust samples tested negative. Herds B and C exhibited high IgG phase I activity, indicating a past infection. In herd B, approximately two-thirds of the goats shed *C. burnetii* with vaginal mucus, and irregular positive results were obtained from BTM. Herd C had two positive goats based on vaginal swabs, and BTM tested positive once. Dust samples from herds B and C contained *C. burnetii* DNA, with higher quantities typically found in samples from the milking parlor. This study highlights the different infection dynamics in three unvaccinated dairy goat herds and the potential use of dust samples as a supportive tool to detect *C. burnetii* at the herd level.

## 1. Introduction

The obligate intracellular, Gram-negative bacterium *Coxiella* (*C.*) *burnetii* is the causative agent of the zoonotic disease Q (query) fever in humans, which is also known as coxiellosis in animals. Except for New Zealand and Antarctica, the pathogen is distributed worldwide [1]. The bacterium exists in two different antigenic variants (phase variation), distinguished by two different forms of lipopolysaccharides (phases I and II), of which phase II variants are exclusively present in laboratory conditions, while phase I variants exist in infected host animals [2]. Many species can transmit *C. burnetii*; however, in Europe, species such as cattle, goats, sheep, water buffaloes, and wild ruminants are the most important reservoirs [3,4,5]. In ruminants, Q fever is often asymptomatic, but can lead to abortions, or stillborn, weak, or premature offspring [6,7]. Abortion rates in goat herds vary, ranging from 5 to 90% [6,8]. Infected goats excrete *C. burnetii* for several weeks via milk, urine, feces, and, in particularly high quantities, with vaginal mucus and placentae [1,9,10,11,12]. Once a goat herd has been infected, *C. burnetii* can persist for long periods. The pathogen maintains itself through persistently infected goats, super-shedding individuals, or the introduction of new susceptible animals into the herd [10,13,14,15].

Spillover transmissions to humans are mostly linked to domestic small ruminants [16,17,18]. In this context, inhalation of contaminated aerosols is the main route of Q fever transmission [1]. The median infectious dose (ID50) for humans has been estimated at 1.5 bacteria, indicating the high infectivity of *C. burnetii* [19]. Approximately 50% of infected humans develop flu-like symptoms such as fever, headache, and pneumonia [20,21]. Approximately 1–5% of cases develop a persistent focalized infection, which is associated with endocarditis, vascular infection, and a possible fatal outcome [1,22,23,24]. Human infection with Q fever by consuming raw milk and raw milk products is a rare event [25,26]. From 2007 to 2011, a seroprevalence in the general human population of between 1.7 and 3.5% was determined in Switzerland [12]. Additionally, the origin of two small-scale human epidemics was traced back to sheep and one small-scale epidemic originated from goats [21,27,28].

In Switzerland, owners of small ruminants are obliged to report all abortions to a veterinarian. Testing for *C. burnetii* is mandatory if there are multiple abortions within 4 months or if the aborting animal is on alpine pastures during summer or in a livestock trade herd (Swiss Ordinance on epizootic diseases, article 129). Between 2012 and 2021, the Swiss Federal Food Safety and Veterinary Office (FSVO) recorded 3–19 cases of coxiellosis in goats each year [28]. In 2011, 3.4% of 321 Swiss goat sera tested positive for *C. burnetii* [12]. Coxiellosis is one of the most common infectious causes of caprine abortion in Switzerland, but *Chlamydia* (*Chl.*) *abortus* is the most frequently detected abortifacient pathogen [29]. However, goat abortions may be under-reported due to extensive husbandry practices and a low tendency among Swiss small ruminant owners to report them, even with incentives [12,30].

Diagnosing *C. burnetii* is challenging, as serologic status and bacterial excretion may not correlate [10,15,30]. The preferred method for antigen detection is real-time PCR, specifically the IS*1111* gene multicopy element assay, which is highly sensitive and a valuable tool for herd surveillance [13,31,32]. Nonetheless, the IS*1111* element is also present in *Coxiella*-like endosymbionts, which occur mainly in ticks and lead to false positive results [33]. Nevertheless, real-time PCR using the IS*1111* multicopy element is suitable for different matrices from ruminants such as vaginal mucus, feces, nasal mucus, individual milk samples, placenta, or abortion material [8,13,30,34]. However, due to variations in IS*1111* multicopies between *C. burnetii* strains, it only provides semiquantitative results [32]. An enzyme-linked immunosorbent assay (ELISA) is recommended for the detection of *C. burnetii* antibodies in ruminants [35]. Commercial ELISA kits detect both IgG phase I (PhI) and phase II (PhII) antibodies simultaneously [16,36]. The differentiation between phase I and phase II antibodies allows for characterizing the disease status in goats and has recently been increasingly used [11,37]. The increase in IgG PhII 3 weeks after infection indicates recent infection, while IgG PhI levels peak 9 weeks after infection and remain elevated for at least 2 years, suggesting a past infection [38,39]. Hence, phase-specific serology is a useful tool for distinguishing infection status in goats [11,37,40].

To identify *C. burnetii*-positive ruminants at the herd level, bulk tank milk (BTM) has been analyzed using PCR [41,42,43]. The shedding of *C. burnetii* in goat milk is, however, intermittent; thus, repeated BTM sampling is required for monitoring [37,44]. Moreover, BTM monitoring is not applicable to non-dairy small ruminants. Recently, dust samples from windowsills and barn facilities have been used as an easy and inexpensive method to detect *C. burnetii* DNA on ruminant farms [37,45,46]. The detection of *C. burnetii* in environmental dust with PCR does not necessarily indicate the actual presence of an infective inhalable agent [47]. In the past, dust samples showed high contamination levels after outbreaks, abortions, and lambing periods, and these correlated well with the infection dynamics of small ruminant herds and remained positive for several months or even until the following lambing season [8,46,47,48].

An inactivated vaccine against *C. burnetii* phase I is licensed for cattle and goats in many European countries, but not in Switzerland. In these countries, dairy goat herds are regularly vaccinated to prevent *C. burnetii* shedding or to control coxiellosis [37,49,50]. Consequently, there are limited data on *C. burnetii* excretion and antibody response in unvaccinated dairy goat herds after natural infection. For instance, phase-specific immune response and pathogen shedding were determined in one unvaccinated dairy goat herd for 3 years [39]. A better understanding of the long-term dynamics of *C. burnetii* infections in such herds is important for prevention and disease monitoring strategies. Therefore, the main focus of this study was to determine the excretion of *C. burnetii* in three unvaccinated dairy goat herds 2 years after the initial identification of the pathogen within the herds. For this purpose, vaginal swabs after parturition and monthly BTM samples were collected and analyzed with PCR. Moreover, monthly collected dust samples from the barn and milking parlor were analyzed to support the findings from the vaginal swabs and BTM samples as an indirect method to monitor *C. burnetii* excretion. A further objective of the study was to characterize possible variations in the immune response against *C. burnetii* during the kidding period. Thus, blood samples were collected at three different sampling dates and analyzed with phase-specific ELISAs.

## 2. Materials and Methods

In 2018/2019, a study investigated *C. burnetii* in caprine abortions in Switzerland using a blended IgG ELISA (Idexx Q Fever, IDEXX Diavet AG, Bäch, Switzerland) for antibody detection and real-time PCR (IS*1111*, TaqMan™ Fast Advanced Master Mix, Applied Biosystems, Inc., Foster City, CA, USA) for *C. burnetii* DNA detection in fetuses and placentae [30]. Three dairy goat herds that tested positive for *C. burnetii* were selected for further investigations. No coxiellosis control measures or vaccination program was implemented in the selected herds. All goats were housed in free-stall barns with permanent or temporary access to a confined outdoor area and pasture. The three herds followed a seasonal kidding scheme, and animal husbandry practices remained unchanged. There was no separate kidding area and the entire herd shared the same air space in the main barn. No specific cleaning measures were implemented after the kidding period (e.g., use of disinfectants). More details on herd parameters and *C. burnetii* history can be found in the Appendix A and in the previous publication [30].

Goats already present in the herd in 2018/2019 were preselected to study the development of coxiellosis 2 years after the first detection of *C. burnetii*. A required sample size of 35 goats per herd was determined (power of 0.95, α = 0.05, G*Power Version 3.1.9.4, Kiel University, Germany). A sonographic pregnancy examination was performed, and 35 randomly selected pregnant individuals from the above-mentioned cohort were chosen from each herd. However, in herds A and B, less than 35 goats fit all these criteria, so only 28 and 29 animals were included in these herds, respectively.

### 2.1. Herd History

#### 2.1.1. Dairy Goat Herd A

Dairy goat herd A (total n = 42) was housed in a building with four solid wooden walls. In January 2019, two abortions were examined, and one doe (#E1) tested positive for *C. burnetii* antibodies, but the abortion material tested negative for *C. burnetii* [30]. However, *C. burnetii* DNA was detected in a single milk sample from goat #E1 4 weeks after the abortion (Cq 38.8). The second doe (#E2) tested negative for *C. burnetii* antibodies, but *C. burnetii* DNA was found in the placenta (Cq 37.8). Unfortunately, this goat died after the initial testing. A total of 28 goats met the requirements described above and were included in the present study. Additionally, two aborted fetuses of goats (#152 and #129) that were not part of the study cohort were analyzed.

The goats were housed on straw bedding, which was completely replaced every 4–5 months. After removing the bedding, neither water nor disinfectants were used. Instead, the floor was covered with lime dust and new straw was added. Milking was conducted directly in the barn, with the goats secured in the feeding fence. The main kidding period occurred from February to March 2021, with two additional goats kidding in April 2021. During the summer months (end of May to September), the goats were moved to alpine pastures. During this period, the winter barn was emptied, cleaned, and washed with water, but no disinfection was performed. Blood sampling in herd A took place over a period of 11 months, from September 2020 to July 2021, due to an extended kidding season.

#### 2.1.2. Dairy Goat Herd B

In dairy goat herd B, a total of 139 animals were housed in a building with three solid wooden walls and one open side. In December 2018, one goat (#C1) that aborted tested positive for *C. burnetii* antibodies, and the pathogen was detected in the placenta (Cq 34.9) [30]. Two weeks after the abortion, *C. burnetii* DNA was found in a fecal sample (Cq 38.0), and four weeks after the abortion, it was detected on a vaginal swab (Cq 36.8) from goat #C1. A total of 29 goats met the aforementioned requirements and were selected for further investigations. Furthermore, two abortions from goats (#565 and #252) that were not part of the study group were analyzed.

The goats were kept on straw bedding, which was completely changed every 1.5–2 months without the use of water or disinfectants. The side-by-side milking parlor was located in a separate building. Some parts of the milking parlor facilities were made of wood, including the elevated milking floor for the goats, which was cleaned with water twice a day. No disinfectants were used in the milking parlor. Sampling took place over an 8-month period (February to September 2021), and kidding occurred in May/June 2021.

#### 2.1.3. Dairy Goat Herd C

Herd C was the largest dairy goat herd included in the study (total n = 224). In December 2018, two goats (#D1 and #D2) experienced abortions. Goat #D1 excreted a high burden of *C. burnetii* (Cq 6) in the placenta, while goat #D2 tested negative for *C. burnetii* in both the placenta and fetus [30]. Both goats had antibodies against *C. burnetii*. Goat #D1 excreted the pathogen via vaginal mucus (Cq 26.9–33.8), feces (Cq 27.3–38.5), and milk (Cq 29.2–33.9) within 3 months after abortion, with measurements taken every 2 weeks. Despite the initial absence of *C. burnetii* in abortion material from goat #D2, the animal shed the pathogen for up to 3 months, with *C. burnetii* detected in vaginal mucus (Cq 28–38.2), feces (Cq 23.3–38), and milk (Cq 29.5–38.9), measured every 2 weeks. From herd C, a total of 35 goats met the aforementioned requirements and were selected for further investigations. Goat #D1 was additionally sampled to the randomly selected study cohort. Two of the studied goats (#20 and #24) showed bloody vaginal discharge but abortion material was not detected. Therefore, only vaginal swabs were tested for *C. burnetii* and *Chl. abortus*. Moreover, abortion material from a non-study goat (#909) was also examined for abortifacient pathogens.

The barn housing of herd C had four solid concrete walls. Straw was used as bedding material, and the bedding was completely replaced every 4 months without using water or disinfectants before applying new straw. Milking took place in a side-by-side milking parlor that shared the same airspace as the main goat barn. The milking parlor had concrete and metal surfaces and was cleaned with water twice daily. No disinfectants were used in the milking parlor. The kidding season for herd C occurred in August 2021, and the sampling period took place over 8 months from April to November 2021.

### 2.2. Collection of Blood Samples and Vaginal Swabs

The *C. burnetii* antibody response of each doe in the study cohort was determined 3 months before the estimated kidding date, within 4 weeks after kidding, and 3 months after kidding. This sampling scheme was designed to determine the serologic status around the kidding period. Serum samples were collected with jugular vein puncture (Monovette 9 mL, Sarstedt AG & Co. KG, Nümbrecht, Germany). The samples were centrifuged at 411× *g* for 10 min, and sera were stored at −20 °C. Additionally, animal owners were instructed to collect a vaginal swab (dry swab 101 × 16.5 mm, Sarstedt AG & Co. KG) from each study goat within 72 h after kidding. Prior to swab collection, the vulva was cleaned using a single-use dry paper towel. Vaginal swabs were stored at −20 °C until further processing. In case of abortion, vaginal swabs and serum samples were obtained following the same procedure as after physiologic kidding.

### 2.3. Collection of Environmental Dust and Bulk Tank Milk

Dust samples from the barn and milking parlor were collected monthly throughout the entire study period. The sampling started 3 months before the expected kidding period and continued until the last blood sample was taken (third sampling date). The timing of the last sampling varied among the herds due to an unexpectedly prolonged kidding season, particularly in herd A. Dry swabs (101 × 16.5 mm, Sarstedt AG & Co. KG) were rolled over a length of 1 m at elevated locations (e.g., windowsills and barn installations) as described elsewhere [8]. These locations were selected to avoid direct animal contact and minimize contamination. Three locations in the barn and three locations in the milking parlor were chosen and documented through photographs to ensure consistent sampling locations throughout the entire study. One-meter-long single-use paper strips were used to measure the collection distance without cross-contamination between sampling locations. The author (C.T.) conducted all dust sampling to ensure a consistent sampling technique. In herd A, there was no separate milking parlor, so three locations near the feeding fence (horizontal surfaces of the feeding fence itself or barn installations directly adjacent to the feeding fence) were classified as the milking parlor dust sampling locations, while three locations further away from the feeding fence (locations approximately 3 m away from the feeding fence) were classified as the barn dust sampling locations. Since the goats were transferred to alpine pastures at the end of May, the dust samples were taken in May and June 2021 from the depopulated barn before cleaning took place.

In addition, BTM samples were collected at the same time as the dust samples (Tube 10 mL, Sarstedt AG & Co. KG). The BTM contained milk from the entire dairy goat herd, and no BTM samples were available during the dry period before kidding. Both the dust and BTM samples were stored at −20 °C until further processing.

### 2.4. Laboratory Analysis of Sera, Vaginal Swabs, Dust Swabs, and Bulk Tank Milk

Goat sera were examined using two phase-specific ELISAs (EUROIMMUN AG, Lübeck, Germany) in accordance with the manufacturer’s instructions, which have been recently described in detail elsewhere [51]. Test results were quantitatively presented in relative units (RU) determined with a standard curve. Serum samples with RU values ≥22 were considered positive. The monthly three dust samples from the barn and milking parlor were pooled for DNA detection. DNA from the vaginal swabs and pooled dust swabs was extracted with an InviMag^®^ Universal Kit/KF96 (STRATEC Molecular GmbH, Berlin, Germany) in accordance with the manufacturer’s instructions using the KingFisher^TM^ Flex (ThermoFisher Scientific GmbH, Dreieich, Germany). Moreover, 2 mL of the BTM was centrifuged for 5 min at 2655× *g*. Subsequently, the fat was removed from the tube using a sterile swab. After another centrifugation step for 10 min at 20,817× *g*, the supernatant was dumped. Bacterial DNA was prepared from the remaining pellet using the InviMag^®^ Universal Kit/KF96 (STRATEC Molecular GmbH, Berlin, Germany) in accordance with the manufacturer’s instructions as well. *C. burnetii*-specific DNA fragments were detected from the swabs and BTM using amplification of IS*1111* elements with a real-time PCR (LSI VetMAX^TM^ *Coxiella burnetii*, Thermo Fisher Scientific GmbH, Dreieich, Germany). The real-time PCR was performed in accordance with the manufacturer’s instructions, and Cq values of ≤45 were considered positive.

### 2.5. Collection and Analysis of Abortion Material

If abortions happened during the study period and were promptly brought to the attention of the study leaders by the animal owners, they were analyzed to determine the presence of abortifacient agents. In herds A and B, two abortions each were reported from animals that were not part of the study population (goat #152 and #129 in herd A, and goat #565 and #252 in herd B). Two goats from the study cohort in herd C exhibited bloody vaginal discharge, but no abortion material was detected. Therefore, only vaginal swabs were analyzed from these goats (#20 and #24). Additionally, abortion material from a goat (#909), which was also not part of the study cohort, was further analyzed.

Examination of the aborted material (placenta and/or fetus) was conducted within 1–2 days after the abortion. The fetus and placenta were examined following a commercial caprine abortion examination protocol provided by a Swiss veterinary pathology institute that adheres to Swiss legal requirements for abortion examination and beyond (Swiss Ordinance on epizootic diseases (TSV), SR.916.401; Article (Art.) 129). It included gross pathology and histology of the fetus and/or placenta; a Stamp’s modified Ziehl–Neelsen staining (presumptive diagnosis for *C. burnetii*, *Chl. abortus*, or *Brucella* spp.); a broad-spectrum culture of the fetal abomasum, liver, and/or lung; and, if available, the placenta as well as special cultures for *Campylobacter fetus* subsp*. fetus, Campylobacter fetus* subsp*. venerealis*, and *Salmonella* spp. as described in detail elsewhere [31]. Additionally, fetal organ pools (the lung, liver, spleen, and, if available, placenta) were prepared as follows: 20 mg of the samples was mixed with 400 µL of molecular-biology-grade water and was crushed with a steel bullet using the TissueLyser^®^ (QIAGEN Benelux B.V., Venlo, The Netherlands) for 2 min at 15 Hz. Afterwards, 200 µL was put in the InviMag^®^ Universal Kit/KF96 (STRATEC Molecular GmbH) and the bacterial DNA was prepared in accordance with the manufacturer’s instructions. Afterwards, *C. burnetii* DNA was detected with real-time PCR as described above, and *Chl. abortus* DNA was identified using real-time PCR in accordance with published protocols [52,53]. Cq values below 38 in the *Chl. abortus* PCR were considered positive. The two vaginal swabs from goats (#20 and #24) in herd C were also examined for *Chl. abortus*.

### 2.6. Statistical Analysis

Statistical analyses were performed using R (R Foundation for Statistical Computing, Vienna, Austria, https://www.R-project.org (accessed on 16 June 2023), and RStudio (Integrated Development for RStudio, Inc., Boston, MA, USA, http://www.rstudio.com (accessed on 8 June 2023). Results with *p* < 0.05 were considered significant.

Vaginal swabs, BTM, and dust swabs were evaluated using descriptive statistics. Direct comparisons of Cq values between herds were avoided because of the semiquantitative nature of the IS*1111* real-time PCR assay. Additionally, vaginal swab results were declared either positive or negative, and herds with positive tests were compared using Fisher’s exact test to compare vaginal shedding among the three herds. IgG levels were checked for normal distribution using Shapiro–Wilk tests and quantile–quantile plots. The nonparametric IgG levels were compared between phase I and phase II at the herd level using Wilcoxon rank-sum tests. Phase I and phase II IgG levels were separately compared between sampling time at the herd level. Since single data points were missing for seven individuals in herd B and one individual in herd A during repeated sampling, linear mixed models were applied for this purpose. In contrast to the non-parametric Wilcoxon test, a comparison with this statistic method remains unbiased regardless of missing repeated values.

## 3. Results

### 3.1. Serology

For the first and second sampling dates, results of the phase-specific serology were obtained from all study animals (herd A: n = 28, herd B: n = 29, and herd C: n = 35). For various reasons (e.g., death), not all goats in herds A and B were available for the third serologic sampling (remaining goats: n = 27 in herd A and n = 23 in herd B). In herd A, *C. burnetii* antibody activities of IgG PhI and PhII did not differ significantly throughout the entire study period and remained below the positivity threshold (Figure 1). A few individuals were seropositive with either both assays or only one ELISA. In herd B, the median IgG PhI levels were more pronounced than the IgG PhII ones, but without statistically significant difference (Figure 1). Only the *C. burnetii* IgG PhI values between pre-kidding and at kidding showed a significant difference, with a higher median level at pre-kidding. In herd C, the *C. burnetii* IgG PhI antibody response was significantly higher than the IgG PhII at all three sampling dates (Figure 1). Moreover, the median of IgG PhI activity pre-kidding was higher than post-kidding (*p* < 0.05), and the IgG PhII response at the second and third sampling dates differed significantly.

### 3.2. Vaginal Swabs

Vaginal swabs were obtained from all study goats (herd A: n = 28, herd B: n = 29, and herd C: n = 35). In herd A, no vaginal swab tested positive for *C. burnetii* DNA. Significantly more goats in herd B (21/29) shed *C. burnetii* with vaginal mucus than goats from herds A and C. Only two goats in herd C excreted the pathogen via vaginal mucus during kidding, and there was no significant difference in positivity rate between herds C and A. More details are illustrated in Figure 2.

### 3.3. Bulk Tank Milk

Due to the dry period in every studied herd, BTM samples were not collected for the entire study period. BTM samples from herd A tested negative for *C. burnetii* DNA during the entire study period (Figure 3). Most BTM samples tested positive during the kidding season and afterwards in herd B. Only one BTM specimen from herd C had detectable *C. burnetii* DNA (Cq 38), and this was obtained before the dry period started.

### 3.4. Dust Samples from Barn and Milking Parlor

Both types of dust samples from herd A tested negative for *C. burnetii* DNA, except for one swab from the milking area (Cq 38). On farm B, all dust swabs from the milking parlor tested positive, with a peak at the beginning of the kidding season. The results from the barn dust sampling showed an undulating trend. Stable levels of *C. burnetii* DNA were obtained from the milking parlor’s dust on farm C, whereas the dust samples from the barn turned negative at the end of the study period. Details are presented in Figure 3.

### 3.5. Abortion Material

All abortions tested negative for *Brucella* spp., *Salmonella* spp., *C. fetus* subsp*. fetus,* and *C. fetus* subsp*. venerealis*. The fetuses showed no macroscopic or microscopic alterations that could justify the abortion. The placentae were unremarkable in macroscopic and microscopic examination, except for two cases. Placentitis was suspected in material from goat #152 (herd A), but a final assessment was hindered due to advanced autolysis. Purulent placentitis with vasculitis was diagnosed in the placenta of goat #565 (herd B). The PCR results for *C. burnetii* and *Chl. abortus* are shown in Table 1.

### 3.6. Vaginal Swab and Phase-Specific Serology from Goat #D1 in Herd C

Goat #D1 aborted in December 2018 due to *C. burnetii* and remained in herd C. Although this goat was not originally included in the randomly selected study cohort, it was also tested in 2021 due to the exceptionally high amount of *C. burnetii* in the placenta (Cq 6) in December 2018. During the kidding season of 2021, goat #D1 successfully gave birth to a living kid. The vaginal swab tested negative for *C. burnetii* DNA, but phase-specific serology revealed positive results at all three sampling dates: pre-kidding: IgG PhI—250.7 RU and IgG PhII—47 RU; at kidding: IgG PhI—188.6 RU and IgG PhII—36.6 RU; and post-kidding: IgG PhI—235.8 RU and IgG PhII—54.7 RU.

## 4. Discussion

The present field study provided new data on the infection dynamics of *C. burnetii* in dairy goat herds where no vaccination program was implemented as a control measure. Such information is crucial for risk assessments for public health authorities and complements findings from herds where vaccination programs were implemented.

Nowadays, the PCR is the preferred method for detecting *C. burnetii* in various sample matrices, and the IS*1111* multicopy element assay is known for its high sensitivity [32]. Rousset and colleagues [54] determined the maximum limit of detection, for the PCR method used (LSI VetMAX^TM^), to be Cq 36.1. Consequently, Cq values above this threshold may be considered inconclusive or even falsely positive, contradicting the manufacturer’s defined threshold of Cq ≤ 45. This consideration should be kept in mind when interpreting the results obtained in this study, particularly when only single positive results were detected, such as the positive milking parlor dust swab (Cq 38) from farm A and the positive BTM sample (Cq 38) from farm C.

The investigation on phase-specific IgG immune responses on a herd level generated useful data for characterizing the stage of *C. burnetii* infection in dairy goat herds [11,37]. Vaginal swabs and dust samples can complement the information provided with phase-specific serology. In this study, the obtained data allowed for determining an infection stage for each of the three studied herds. In herd A, several goats tested positive for IgG PhI and/or PhII antibodies, but the median antibody levels remained below the ELISA threshold. Considering the *C. burnetii* results from 2018 (placenta, Cq 37.8; milk, Cq 38.8) and the negative outcomes for *C. burnetii* in the vaginal swabs, BTM samples, and abortion material from 2021, it appears that *C. burnetii* induced an immune response in goats but did not result in pathogen excretion in 2021. This observation is supported with the absence of *C. burnetii* DNA in almost all dust samples from the barn and milking parlor. Consequently, herd A appears not to have experienced a coxiellosis outbreak. In herd B, the study cohort showed elevated levels of both phase-specific antibody levels with a higher IgG PhI activity, but the medians of IgG PhI and IgG PhII did not statistically differ. According to previous findings [37] and in the context of the high number of goats shedding *C. burnetii* via vaginal mucus, the infection status can be interpreted as an ongoing infection. Furthermore, both abortions in herd B tested positive for *C. burnetii*, and the dust samples from the facilities contained *C. burnetii* DNA, with the milking parlor dust showing a peak at the beginning of the kidding season. The IgG PhI response in study cohort C was significantly higher than the IgG PhII response at all three sampling dates. This suggests that the herd had successfully overcome the infection and developed immunity, resulting in reduced shedding of the pathogen via vaginal mucus and milk, as described elsewhere [39]. Furthermore, the follow-up investigation of high-shedding goat #D1 in 2018 supports this interpretation of the *C. burnetii* infection status in 2021. However, the dust samples from the barn and milking parlor remained positive for *C. burnetii* throughout almost the entire study period.

The serologic sampling scheme compared the antibody levels at different stages of reproduction and revealed deeper insights in the immune response around the kidding season. The IgG PhII response peaked in flocks B and C at the time of kidding, with median values in flock C significantly higher at parturition than 3 months later. This effect was not observed for PhI antibodies, and median levels were even significantly lower in herd B at the time of kidding than before. Nevertheless, the increase in IgG PhII in two of three herds may be related to the influence of pregnancy hormones, progesterone, and estrogen, on the immune response against *C. burnetii* [55,56]. The phenomenon of an increasing immune response against *C. burnetii* in goats shortly before or after parturition has been discussed previously, and it has been suggested that increasing estradiol levels and a sudden drop in progesterone at the end of gestation stimulate the immune response against *C. burnetii* [38,57]. The effect on PhII antibodies alone remains unclear, and studies under controlled conditions are urgently needed to clarify the complex issue of *C. burnetii* immunity and hormones around parturition to improve our understanding of Coxiella pathogenicity.

Overall, the results of the serologic testing and the detection of the pathogen in vaginal swabs and dust samples 2 years after the initial detection of *C. burnetii* show considerable variations among the three dairy goat herds. In particular, the differences between herds B and C raise questions, as both herds had goats shedding the pathogen for several weeks. It is possible that herd size and different management practices, such as differences in the duration of the kidding season (see Appendix A), contributed to these divergent outcomes. Between 2018 and 2021, the owner of herd B purchased several new female goats from various sources, representing a potential risk factor for introducing *C. burnetii* into small ruminant herds and sustaining the circulation of *C. burnetii* [58]. In contrast, the farmer of herd C purchased only a few animals from farms with advanced biosecurity standards. Nevertheless, even in a herd without newly introduced goats, a high number of vaginal shedders were identified in two consecutive parturitions [10]. Interestingly, herd A did not show an ongoing infection although the goats were mixed with other herds on summer pastures, which possibly is related to the fact that usually no kidding takes place in this period and only a minor influence of alpine grazing on the seroprevalence of other abortifacient agents has been demonstrated [59].

In recent years, the use of dust samples as a method to detect *C. burnetii*-positive livestock has gained popularity, but validation of this new diagnostic approach is still lacking [46,48,60]. In this study, the detection of *C. burnetii* in environmental dust samples was used as an additional sampling approach to determine the presence of *C. burnetii* shedding in goat herds. In agreement with previous studies [37,45], a correlation between the vaginal shedding and *C. burnetii* detection in dust samples was found. In herds B and C, animals shedding *C. burnetii* in vaginal mucus were detected, and dust samples from the barn and milking parlor tested positive multiple times. In contrast, in herd A, where no shedding animals were present, all dust samples were negative except for a single swab. Higher levels of *C. burnetii* DNA were detected in dust swabs from the milking parlor compared to the samples from the barn. These findings are consistent with a previous study [37], thus highlighting the significant contamination of milking parlors in *C. burnetii*-positive dairy goat farms. In the past, this facility unit was neglected in risk assessments despite the known risk of humans acquiring Q fever during milking activities [61]. Based on our findings, we conclude that milking parlors should be the favored sampling locations.

The results of the dust samples from the barn and milking parlor in this study raise questions about the viability of *C. burnetii* detected in dust samples and the duration of environmental contamination. DNA detection alone does not provide information about infectiousness, but it is assumed that *C. burnetii* remains contagious for up to 2 months after parturition [48]. However, evaluating the bacterial viability in environmental specimens is challenging, still requires inoculation into live animals, and is a time- and labor-intensive process [47,48,62,63]. Consequently, techniques to assess viability are not suitable for routine diagnostics. Additionally, only samples containing high quantities of *C. burnetii* (Cq < 30) are appropriate for viability evaluation [60]. Therefore, methods to determine the risk of *C. burnetii* contamination in animal husbandries and milking parlors are urgently needed. The lack of such methods hampers the Q fever risk assessment for farmers, farm workers, and individuals living near *C. burnetii*-positive farms.

The outcome of the investigation of dust samples depends on several factors, including the sampling method, sampling location, PCR method, activities during sampling, history of abortion, number of reproductive females, number of vaginal shedders, ruminant species, and cleaning/disinfection measures [45,47,60,64,65,66]. Without cleaning and disinfection, *C. burnetii* DNA can be detected for several months in dust from farm facilities [48,66], which is consistent with the findings of this study.

Finally, the use of dust samples as a monitoring tool requires further validation in the future, and reliable thresholds need to be established to characterize the disease status at the herd level. On the other hand, repeated negative results of dust samples in barns confirm the absence of the pathogen in housed ruminant herds.

Overall, there was a strong correlation between vaginal excretion and milk shedding in herds A und C. However, in herd B, there was one negative BTM sample despite the presence of several goats shedding the pathogen through vaginal mucus. This discrepancy could be attributed to intermittent milk excretion [44]. This highlights the necessity for repeated BTM sampling to avoid false negative samples [41].

In the present study, *Chl. abortus* was detected in abortion material in two goat herds (A and B). Additionally, a co-infection was detected with *C. burnetii* in herd B. Co-infections with both pathogens appear to occur regularly [3,31,67,68]. Therefore, investigation for multiple abortifacient agents remains crucial even after the detection of *C. burnetii*.

The authors are aware of the limitations of the present field study. Our main objective was to investigate the disease development without any countermeasures in multiparous goats that had been present in the herds since 2018, thus providing an update on this important issue [39]. Therefore, primiparous and purchased animals were not considered although they can maintain or introduce the *C. burnetii* infection into a herd [15,58]. Extrapolations to large, intensively managed dairy goat herds should be made with caution, as the herds studied were small compared to dairy goat herds in other countries. The potential interactions between different species on the same farm were not explored, although they can have significant implications [57]. Nevertheless, this study provides valuable insights into the zoonotic risk associated with naturally *C. burnetii* infected and unvaccinated dairy goat herds in Switzerland.

## 5. Conclusions

The current field study updated our knowledge of the infection dynamics of *C. burnetii* in unvaccinated dairy goat herds [39] and showed that two out of three herds shed *C. burnetii* 2 years after the initial pathogen detection. Phase-specific serology seems to be a valuable tool to gain deeper insights into the disease status of dairy goat herds affected by *C. burnetii*. These data can be used for further risk assessments. The results of the molecular investigations showed a correlation between the presence of shedding animals and environmental contamination, which needs further investigation. Particularly, dust samples from milking parlors appeared to be a simple and cost-effective tool for detecting *C. burnetii*-positive dairy goat farms, regardless of their lactation status. Overall, *C. burnetii* was found to be shed in two of three dairy goat herds, even 2 years after the initial detection, and the facilities remained contaminated. Based on these findings, it is suggested to implement effective measures, such as animal vaccination and the cleansing and disinfection of facilities, in order to prevent a possible transmission to humans.

## Figures and Tables

**Figure 1 animals-13-03048-f001:**
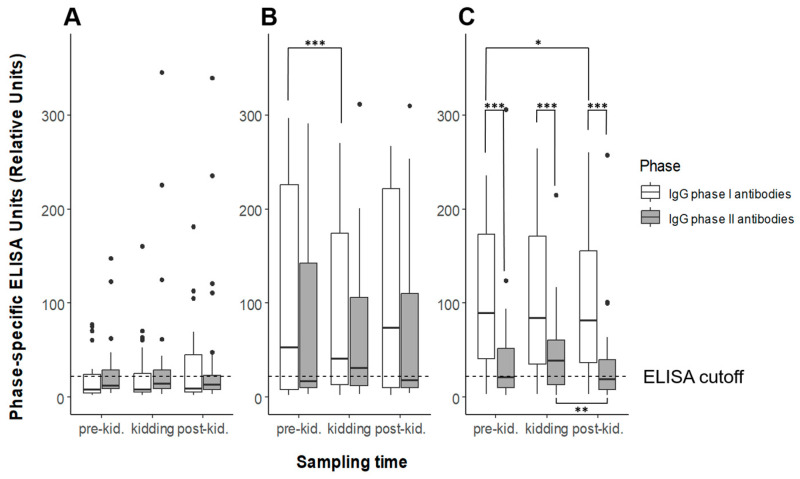
Antibody levels of *Coxiella burnetii* IgG phase I (white) and phase II (gray) detected with phase-specific ELISAs in three dairy goat herds ((**A**–**C**) from left to right) during three sampling dates (pre-kidding, kidding, and post-kidding). Outliers are presented as dots; the black bar marks the median. Positivity threshold: RU ≥ 22. Significant differences between medians are indicated with asterisks: * *p* < 0.05, ** *p* < 0.01, *** *p* < 0.001.

**Figure 2 animals-13-03048-f002:**
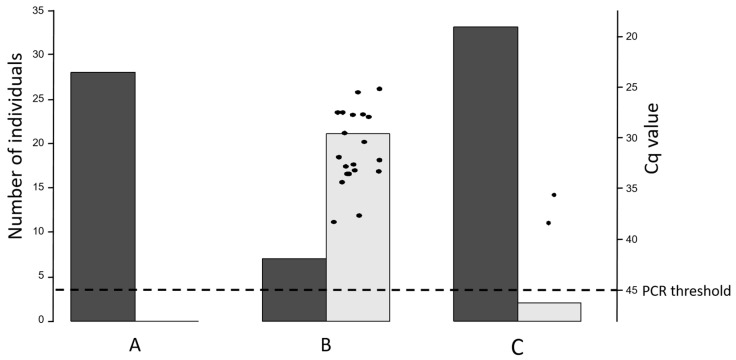
*Coxiella burnetii* shedding through vaginal mucus at kidding in three dairy goat herds (**A**–**C**) analyzed with real-time PCR. The left *y*-axis indicates the numbers of negative (dark-gray column) and positive (light-gray column) vaginal swabs. The right *y*-axis and the individual dots demonstrate the Cq values of the positive vaginal swabs. Positivity threshold ≤45 Cq.

**Figure 3 animals-13-03048-f003:**
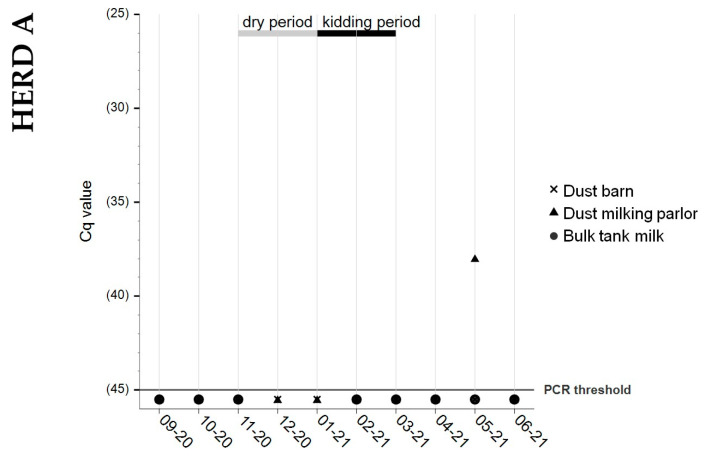
*Coxiella burnetii* shedding detection with real-time PCR in monthly collected samples of **✕** barn dust, ▲ milking parlor dust, and ● BTMin herds A, B, and C. Sample dates indicate month/year of sampling. No dust and BTM samples were available for July 2021 from herd B. No bulk tank milk samples were available in herd A for December 2020 and January 2021; in herd B for March and April 2021; and in herd C for June 2021 (dry period; gray bar). Black bar: kidding period. Positivity threshold: Cq ≤ 45.

**Table 1 animals-13-03048-t001:** Results of aborted fetuses and/or placentae analyzed with real-time PCR for *Coxiella burnetii* and *Chlamydia abortus*. * Goats showed bloody vaginal discharge, but no abortion material was detected.

Animal ID	PCR Result: *C. burnetii* (Material, Cq Value)	PCR Result: *Chl. abortus* (Material, Cq Value)
**Herd A**		
Goat #152 (primiparous, purchased)	negative (fetus and placenta)	negative (fetus and placenta)
Goat #129 (primiparous)	negative (fetus)	positive (fetus, Cq 36.9)
**Herd B**		
Goat #565 (purchased)	positive (placenta, Cq 39.3) negative (fetus)	positive (fetus, Cq 36.9; placenta, Cq 27.6)
Goat #252 (purchased)	positive (placenta, Cq 38.4) negative (fetus)	negative (fetus and placenta)
**Herd C**		
Goat #20 (multiparous)	negative (vaginal swab *)	negative (vaginal swab *)
Goat #24 (multiparous)	negative (vaginal swab *)	negative (vaginal swab *)
Goat #909	negative (fetus)	negative (fetus)

## Data Availability

The datasets generated and/or analyzed during the current study are not publicly available due to the individual privacy of the goat owners, but are available from the corresponding author on reasonable request.

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
