# Peer review of "Two Years after Coxiella burnetii Detection: Pathogen Shedding and Phase-Specific Antibody Response in Three Dairy Goat Herds"

_animals, 2023, doi:10.3390/ani13193048_

Round 1

Reviewer 1 Report

"Two years after Coxiella burnetii detection: pathogen shedding and phase-specific antibody response in three dairy goat herds" is a well written manuscript based on a solid study design and interesting outcome for rural flocks of small to medium size. 

I suggest several minor revisions to improve the quality and understandability of this work. 

1. I understand that several abortions belonging to goats that were not selected for the study were submitted and analyzed as well. It would be interesting to know about this while reading the overall information on herd history (Section 2.1). E.g.: add between Lines 164-165 a sentence similar to "xx number of fetuses of does not included in the study were analyzed" and the same for all dairy goat herds. 

2. Line 251 - add "." between the citation and the new sentence.

3. Line 488-491: "Inoculated mice... growth rate determination" - information not pertinent for this work. It would suffice stating that proving viability would request inoculation in live animals as authors stated shortly after. I suggest eliminating the mentioned sentences. 

4. The conclusion section shows some weaknesses. Three herds were taken into consideration, which did not perform any type of sanification. How do you know what the outcome ("These findings emphasize the importance of implementing effective measures, ..., in order to prevent transmission to humans") would have been if there is no comparison with other management types? Is it possible conducting the study on one more herd (with disinfecting practices)? If not, I suggest adding a short comparison with other literature works. 

Thank You.

Author Response

Answers to reviewer’s comments

Reviewer 1

Thank you for taking the time to review our manuscript and providing constructive comments.  Your proposals improved the quality of the manuscript.

  1. I understand that several abortions belonging to goats that were not selected for the study were submitted and analyzed as well. It would be interesting to know about this while reading the overall information on herd history (Section 2.1). E.g.: add between Lines 164-165 a sentence similar to "xx number of fetuses of does not included in the study were analyzed" and the same for all dairy goat herds.

Thank you for this valuable comment. In order to improve the comprehensionof the manuscript, we added a sentence similar to the one you proposed in the sections describing herds A-C as you suggested.

  1. Line 251 - add "." between the citation and the new sentence.

The missing punctuation was added as suggested.

  1. Line 488-491: "Inoculated mice... growth rate determination" - information not pertinent for this work. It would suffice stating that proving viability would request inoculation in live animals as authors stated shortly after. I suggest eliminating the mentioned sentences.

We agree that this information is not crucial for understanding the study and therefore we removed these sentences as suggested.

  1. The conclusion section shows some weaknesses. Three herds were taken into consideration, which did not perform any type of sanification. How do you know what the outcome ("These findings emphasize the importance of implementing effective measures, ..., in order to prevent transmission to humans") would have been if there is no comparison with other management types? Is it possible conducting the study on one more herd (with disinfecting practices)? If not, I suggest adding a short comparison with other literature works.

Thank you for the excellent crosscheck. Unfortunately, we have no access to an additional herd that implemented disinfection measures that we can study in a similar time period after the detection of C. burnetii. Based on our findings from herds B and C, which still shed C. burnetii, we recommend a vaccination program since it has been proven by multiple studies as an effective measure to control C. burnetii (Rousset et al. 2009, DOI: 10.1111/j.1469-0691.2008.02220.x; Hogerwerf et al. 2011, DOI: 10.3201/eid1703.101157; de Cremoux et al. 2012,  DOI: 10.1111/j.1574-695X.2011.00892.x). Moreover, dust samples still contained C. burnetii DNA and therefore, cleaning and disinfection may reduce the contamination of the facilities. To our best knowledge, we are not aware of any studies which used cleansing and disinfection as the only control measure in dairy goat herds. Nevertheless, we revised the sentence and weakened our statement: “Based on these findings, it is suggested to implement effective measures, such as animal vaccination and the cleansing and disinfection of facilities, in order to prevent a possible transmission to humans.

Reviewer 2 Report

The paper "Two years after Coxiella burnetii detection: pathogen shedding 2 and phase-specific antibody response in three dairy goat herds" by Christa Trachsel et al. describe the kinetics of Coxiella burnetii infection in unvaccinated dairy goats.

Although stated at the end of the paper, since the idea was to assess the infection kinetics, the work's significant flaw is not to evaluate primiparous animals. Nevertheless, it is a good work.

Additionally, the manuscript would benefit from English editing.

Regarding the paper and its sections, I would like to raise a few observations and questions:

Simple summary – line 19: here, and throughout the article, the term "chronic Q fever" was used. There is a consensus in recent publications that the correct designation should be "persistent focalized infection" once the clinical picture of this condition is diverse and related to the organ/tissue affected. Please, revise the manuscript. (doi: 10.1080/14787210.2020.1699055)

Introduction – lines 61 to 62: it is essential to emphasize that, although there is an immunological response to different LPS phases, in field conditions, we only have phase I bacteria. Phase II bacteria is only achieved under laboratory conditions.

Introduction – line 63: ruminants are considered the main reservoir of human infection. Virtually many animals can serve as reservoirs of Coxiella, and the epidemiological picture will vary according to regions (see French Guiana).

Introduction – line 76: change the term "chronic infection" to persistent focalized infection

Introduction – lines 96 to 102: Once there are studies indicating the detection of Coxiella-like in animals, it is important to highlight that the IS1111 S1111 is not specific to C. burnetii, and detection assays based only on this element may lead to misidentification with Coxiella-like endosymbionts. (see DOI: 10.1093/femsle/fnv132 and 10.1371/journal.pone.0156710)

Introduction – line 130: "using new diagnostic tools". Nothing new here. You are using PCR and serology.

Materials and Methods – lines 147 to 149: It is known the importance of young animals for the maintenance and dispersion of Coxiella. Why not sample a group of young animals? (primiparous). This approach would give the authors insights regarding infection status and Coxiella circulation.

Materials and Methods – lines 238 to 240: "three locations near the feeding fence were classified as the milking parlor dust sampling locations, while three locations further away from the feeding fence were classified as the barn dust sampling locations". Please, classify "near" and "further away" in meters (or any suitable distance measure).

Materials and Methods – line 251: a missing period. (Throughout the manuscript, there are missing punctuations)

Materials and Methods – Lines 285 to 286: these agents are known to reach the fetus orally. Why was PCR not performed on the stomach contents? It would be an important sample for detecting Coxiella and other agents.

Discussion – Lines 406 to 409: In addition to the information provided, it is also worth mentioning that the manufacturer does not describe the use of the kit for dust analysis. Is there any study validating this kit for this kind of sample?

Discussion – Lines 417 to 419: Additionally, this is the expected infection kinetics, where a herd with reproductive problems in one year if it does not have the insertion and new animals, will present reduced excretion and absence of abortions. Similar to the vaccination effect.

Discussion – Line 422: "outbreak, and Chl. abortus might be the main abortifacient agent in this herd." There is no data for this observation. A single PCR in a fetus is not enough.

Discussion – Lines 423 to 427: "In herd B, the study cohort showed a remarkable IgG PhI activity compared to the IgG PhII activity at all three sampling dates, but these differences were not statistically significant. According to previous findings [30] and in the context of the high number of goats shedding C. burnetii via vaginal mucus, the infection status can be interpreted as an ongoing infection.

The assertion of an ongoing infection should be supported by the rise in phase I and II titers in a period. Moreover, phase II would also rise during the reactivation of persistent infection (not only phase I). Graph 1 shows animals with much higher Phase II titers in herd B than in the others (despite the average being similar).

Moreover, the authors' statement does not corroborate the cited work of Bauer et al., which is as follows: "The rise in IgG PhII antibodies without the appearance of IgG PhI is interpreted as a recently acquired infection. EQUAL LEVELS OF IGG PhI AND PhII REPRESENT AN ONGOING INFECTION AND OCCUR APPROXIMATELY NINE WEEKS POST-INFECTION. Exclusive presence of IgG PhI antibodies outlined an infection in the past".

So, the only technique that gives the idea of ongoing infection is PCR detection.

Discussion – Lines 456 to 459: "Between 2018 and 2021, the owner of herd B purchased several new female goats from various sources, representing a potential risk factor for introducing C. burnetii in small ruminant herds and sustaining the circulation of C. burnetii".

The lack of primiparous and multiparous groups in this study is a weak point, especially with this paragraph. A statement explaining why this evaluation wasn't done would be beneficial.

Overall, the paper is well written, but there are some mistakes regarding punctuation and grammar.

Author Response

Answers to reviewer’s comments

Reviewer 2

The paper "Two years after Coxiella burnetii detection: pathogen shedding 2 and phase-specific antibody response in three dairy goat herds" by Christa Trachsel et al. describe the kinetics of Coxiella burnetii infection in unvaccinated dairy goats.

Although stated at the end of the paper, since the idea was to assess the infection kinetics, the work's significant flaw is not to evaluate primiparous animals. Nevertheless, it is a good work.

Additionally, the manuscript would benefit from English editing.

Regarding the paper and its sections, I would like to raise a few observations and questions:

Simple summary – line 19: here, and throughout the article, the term "chronic Q fever" was used. There is a consensus in recent publications that the correct designation should be "persistent focalized infection" once the clinical picture of this condition is diverse and related to the organ/tissue affected. Please, revise the manuscript. (doi: 10.1080/14787210.2020.1699055)

Thank you for addressing this matter of nomenclature. We revised the suggested manuscript and implemented the term “persistent focalized infection” instead of the formerly used “chronic Q fever” throughout our manuscript. Considering recommendations from other reviewers, we rephrased the sentence as follows: “Inhalation of contaminated aerosols is the main route of transmission to humans in which C. burnetiii can cause a persistent focalized infection with severe consequences.”

Introduction – lines 61 to 62: it is essential to emphasize that, although there is an immunological response to different LPS phases, in field conditions, we only have phase I bacteria. Phase II bacteria is only achieved under laboratory conditions.

Thank you for pointing out the lack of this essential information. We added this information to the sentence: “The bacterium exists in two different antigenic variants (phase variation), distinguished by two different forms of lipopolysaccharides (phases I and II), of which phase II variants are exclusively present in laboratory conditions, while phase I variants exist in infected host animals.”

Introduction – line 63: ruminants are considered the main reservoir of human infection. Virtually many animals can serve as reservoirs of Coxiella, and the epidemiological picture will vary according to regions (see French Guiana).

We agree that information about a broader spectrum of potentially transmitting species other than ruminants is important. Thank you for pointing out the special reservoir and epidemiological situation in French Guiana to us, however, in the studied continent (Europe), ruminants are crucial transmitters of Q fever. Thus, we rephrased the sentence while also embedding suggestions from other reviewers: “Many species can transmit C. burnetii; however, in Europe, species such as cattle, goats, sheep, water buffaloes, and wild ruminants are the most important reservoirs.”

Introduction – line 76: change the term "chronic infection" to persistent focalized infection

We changed the term as suggested throughout the manuscript.

Introduction – lines 96 to 102: Once there are studies indicating the detection of Coxiella-like in animals, it is important to highlight that the IS1111 S1111 is not specific to C. burnetii, and detection assays based only on this element may lead to misidentification with Coxiella-like endosymbionts. (see DOI: 10.1093/femsle/fnv132 and 10.1371/journal.pone.0156710)

Thank you for your input. We added a phrase focused on this fact as well as the recommended literature reference and slightly changed the following phrase to maintain comprehensibility. It reads as follows: “Nonetheless, the IS1111element is also present in Coxiella-like endosymbionts, which occur mainly in ticks and lead to false positive results. Nevertheless, real-time PCR using the IS1111 multicopy element is suitable for different matrices from ruminants such as vaginal mucus, feces, nasal mucus, individual milk samples, placenta, or abortion material” 

Introduction – line 130: "using new diagnostic tools". Nothing new here. You are using PCR and serology.

We deleted “using new diagnostic tools”.

Materials and Methods – lines 147 to 149: It is known the importance of young animals for the maintenance and dispersion of Coxiella. Why not sample a group of young animals? (primiparous). This approach would give the authors insights regarding infection status and Coxiella circulation.

Thank you for your comment. We covered this matter in the reply to your comment concerning lines 456 to 459 of the manuscript.

Materials and Methods – lines 238 to 240: "three locations near the feeding fence were classified as the milking parlor dust sampling locations, while three locations further away from the feeding fence were classified as the barn dust sampling locations". Please, classify "near" and "further away" in meters (or any suitable distance measure).

Thank you for pointing out this missing information. We specified this information as proposed. We would like to emphasize that the sampled barn housed 42 animals only, thus, the eligible sampling sites classified as “barn” had a limited distance from the feeding fence.

Materials and Methods – line 251: a missing period. (Throughout the manuscript, there are missing punctuations)

We added the missing punctuation.

Materials and Methods – Lines 285 to 286: these agents are known to reach the fetus orally. Why was PCR not performed on the stomach contents? It would be an important sample for detecting Coxiella and other agents.

Thank you for the thorough crosscheck. Abortion analysis was not the main focus of the study and rather served as additional information for the other investigations done in the herds. C. burnetii and Chl. abortus replicate in the trophoblasts. Thus, we consider the placenta as the most important sampling tissue to identify both pathogens and C. burnetii was isolated more frequently from the placenta than from fetal material in ruminants (Jones et al. 2010; DOI: 10.1136/vr.c4040). For the detection of the other pathogens, a broad-spectrum culture and special cultures were performed on stomach content as described in the manuscript.

Discussion – Lines 406 to 409: In addition to the information provided, it is also worth mentioning that the manufacturer does not describe the use of the kit for dust analysis. Is there any study validating this kit for this kind of sample?

Thank you for bringing our attention to this point because in fact, the manufacturer does not describe dust as a validated sample matrix. To the best of our knowledge, there are no studies available that validated this kit for dust samples. Nevertheless, this sample matrix has been analyzed using the same kit previously to detect C. burnetii DNA in dust samples (Bauer et al. 2022, DOI: 10.3390/vetsci9030102) and multiple studies used PCR methods with the same target (IS1111) (Zendoia et al. 2021, DOI: 10.1111/zph.12871; Àlvarez-Alonso 2020, DOI: 10.3389/fvets.2020.00352; Hogerwerf et al. 2012, DOI: 10.1128/AEM.00677-12). We chose this kit because it is validated both for milk and vaginal swabs which were also analyzed in our study and avoiding the use of multiple PCR kits was a priority since otherwise, the results may be hardly comparable. We would also like to point out that in general, dust has been evaluated as a matrix that can be reliably screened for C. burnetii by PCR (De Bruin et al. 2011, DOI: 10.1128/AEM.05097-11) and that in our paper the need for standardized protocols for dust sample analysis is sufficiently stated in the discussion section.

Discussion – Lines 417 to 419: Additionally, this is the expected infection kinetics, where a herd with reproductive problems in one year if it does not have the insertion and new animals, will present reduced excretion and absence of abortions. Similar to the vaccination effect.

Thank you for your comment and we appreciate your input.

Discussion – Line 422: “outbreak, and Chl. Abortus might be the main abortifacient agent in this herd.” There is no data for this observation. A single PCR in a fetus is not enough.

Thank you for the excellent crosscheck, we deleted this part of the sentence.

Discussion – Lines 423 to 427: "In herd B, the study cohort showed a remarkable IgG PhI activity compared to the IgG PhII activity at all three sampling dates, but these differences were not statistically significant. According to previous findings [30] and in the context of the high number of goats shedding C. burnetii via vaginal mucus, the infection status can be interpreted as an ongoing infection.

The assertion of an ongoing infection should be supported by the rise in phase I and II titers in a period. Moreover, phase II would also rise during the reactivation of persistent infection (not only phase I). Graph 1 shows animals with much higher Phase II titers in herd B than in the others (despite the average being similar).

Moreover, the authors' statement does not corroborate the cited work of Bauer et al., which is as follows: "The rise in IgG PhII antibodies without the appearance of IgG PhI is interpreted as a recently acquired infection. EQUAL LEVELS OF IGG PhI AND PhII REPRESENT AN ONGOING INFECTION AND OCCUR APPROXIMATELY NINE WEEKS POST-INFECTION. Exclusive presence of IgG PhI antibodies outlined an infection in the past".

So, the only technique that gives the idea of ongoing infection is PCR detection.

We appreciate this comment, and we would like to clarify this issue. In herd B, the study cohort showed elevated levels of both phase-specific antibody levels with a higher IgG PhI activity, but the medians of IgG PhI and IgG PhII did not statistically differ. According to previous findings [35] and in the context of the high number of goats shedding C. burnetii via vaginal mucus, the infection status can be interpreted as an ongoing infection. The combination of vaginal swabs and serology on a herd basis is crucial to determine the infection status as “ongoing”.  This infection status (interpreted as ongoing) is in accordance with the findings in the study of Bauer et al. 2022; DOI: 10.3390/vetsci9030102 in herd B (please note that by coincidence, both our study and the study of Bauer et al. 2022 included a study group that was named herd B). Bauer et al. 2022 equally used vaginal swabs and serology on a herd level to support the declaration of an ongoing infection status. Moreover, these results are what the citation of the work of Bauer et al. 2022 in the discussion section of our manuscript refers to. It does not refer to the sentence “EQUAL LEVELS OF IGG PhI AND PhII REPRESENT AN ONGOING INFECTION AND OCCUR APPROXIMATELY NINE WEEKS POST-INFECTION”, which is just a citation of the study of Roest et al. 2013; DOI: 10.1186/1297-9716-44-67that has been included in the introduction section of the study of Bauer et al. 2022.

Concerning the time period which is needed as a parameter to support a herd infection status as ongoing as it has been brought up by you, to the best of our knowledge, currently it is known that nine weeks are the point at which the start of an ongoing infection has been detected so far on an individual animal level (Roest et al. 2013; DOI: 10.1186/1297-9716-44-67). So far, on a herd level no end point of an ongoing infection has been determined. Nevertheless, in order to improve the comprehensibility of the sentence, we rephrased it as follows: “In herd B, the study cohort showed elevated levels of both phase-specific antibody levels with a higher IgG PhI activity, but the medians of IgG PhI and IgG PhII did not statistically differ.”

Discussion – Lines 456 to 459: "Between 2018 and 2021, the owner of herd B purchased several new female goats from various sources, representing a potential risk factor for introducing C. burnetii in small ruminant herds and sustaining the circulation of C. burnetii".

The lack of primiparous and multiparous groups in this study is a weak point, especially with this paragraph. A statement explaining why this evaluation wasn't done would be beneficial.

We are aware of the importance of primiparous animals in C. burnetii infection dynamics and different authors already described the shedding behavior of primiparous goats in C. burnetii positive herds (de Cremoux et al. 2012, DOI: 10.1111/j.1574-695X.2011.00893.x; Rousset et al. 2009, DOI: 10.1111/j.1469-0691.2008.02220.x; Hogerwerf et al. 2011, DOI: 10.3201/eid1703.101157). Nevertheless, we also agree that sampling this additional animal group would have given even more details. However, we decided to focus on multiparous animals that were already present during the first detection. Our aim was to investigate the persistence and infection dynamics in multiparous animals which were exposed to the pathogen in order to update and extend the findings which were obtained for 20 years ago (Hachette et al. 2003, DOI: 10.1089/153036603765627415). Moreover, the sizes in the investigated herds are considerably small, thus forming primiparous study groups fulfilling statistical criteria would have been impossible or even challenging in this baseline study.

Also, we are aware that introduction of new animals is a risk factor for introducing and sustaining C. burnetii in small ruminant herds. Unfortunately, the number of eligible herds was very limited since it was depending on cases found in a previous study (Heinzelmann et al. 2020; DOI: 10.17236/sat00275). In general, in Switzerland it is not possible to find dairy goat herds without any animal movements and biosecurity standards are traditionally low compared to other countries. Moreover, a challenge was to find animal owners that agreed to participate in the study since many owners hesitate to participate in investigations on diseases with potential legal consequences (C. burnetii is regulated by the Swiss ordinance on epizootic disease). Thus, we had to include such a herd in our study and we nevertheless considered it a valuable and rare opportunity to investigate the dynamics of C. burnetii two years after initial detection.

However, to clarify the reasons for not including primiparous goats in the study, we rephrased a part of the last paragraph of the discussion section as follows: “The authors are aware of the limitations of the present field study. Our main objective was to investigate the disease development without any countermeasures in multiparous goats that had been present in the herds since 2018, thus providing an update on this important issue. Therefore, primiparous and purchased animals were not considered although they can maintain or introduce the C. burnetii infection into a herd.”

Comments on the Quality of English Language: Overall, the paper is well written, but there are some mistakes regarding punctuation and grammar.

Your feedback concerning the quality of English is much appreciated. Upon your recommendation, punctuation, grammar, and other linguistic aspects have been revised by a native-speaker.

Reviewer 3 Report

The work presented by Trachsel et al. and entitled "Two years after Coxiella burnetii detection: pathogen shedding and phase-specific antibody response in three dairy goat herds" is a work describing the effects of the circulation of Coxiella burnetii on 3 goat farms in Switzerland (not vaccinated). The animals are characterized both serologically (using a phase-specific ELISA) and molecularly (individual swabs and bulk milk), thus evaluating the excretion and seroconversion of the animals two years after certain outbreaks. Even if it takes into consideration only three farms, the work in general is constructed in an appropriate manner provide novelty to literature. The authors also appropriately discuss the limitations of their work. Certainly, some sections of the work (the introduction and discussion) present a lot of superfluous information outside the scope of the work, which the authors should eliminate to facilitate readability. I also believe that a round of review is needed before the manuscript is accepted. Below are my specific comments.

Lines 18–20: This sentence is misleading. The authors could limit themselves to mentioning Q fever as a zoonosis.

Line 21: The authors wrote "In a previous study, C. burnetii was detected in three dairy goat herds, but no control measures were implemented". This sentence is not necessary in the "simple summary" section.

Line 27: "past infection". The presence of phase I antibodies could also indicate a chronic infection, not just a past infection.

Line 29: The authors wrote "And dust swabs from the milking parlor can serve as an easy diagnostic tool". I don't agree with this sentence because, although molecular biology applied to unconventional samples can give additional information in Q fever outbreaks, it is not possible to consider it a "diagnostic tool".

Line 32: "pathogen detection"? Maybe it would be better to convert it to outbreak."

Line 43: The final sentence of the abstract could best summarize what was done and the impact of the new information obtained.

Introduction:

Line 63: There is no reference to the exposure of the various ruminant species in Europe. In Europe, infection is present in cattle, small ruminants, buffaloes, and wild ruminants. I recommend this work to mention the presence of the infection also in the Mediterranean buffalo. doi: 10.3390/pathogens11080901

Line 77: A reference to the Dutch outbreak, which, by the way, was mainly caused by goats, is needed. doi: 10.1016/j.medmal.2014.02.006

Line 104: I also recommend another reference for this sentence. doi: 10.1177/10406387221093581.

Some information in the Introduction should be streamlined and summarized.

Material and methods:

Line 135: I’m not sure that "blended" is the appropriate term.

Line 137: "antigen". With the PCR described by the authors, they did not find the Coxiella antigen but its DNA.

Line 146: Do you have data on the productivity of these companies? Is it possible to carry out some kind of statistical analysis to highlight changes in productivity starting before the outbreak, during the outbreak, and in the following two years?

Line 318: I think the authors should delete "eartag loss".

Figure 1: This figure needs better definition or a better graphic design. A different graphic layout should also be taken into consideration for Figure 2 (for example, the cut-off line is not very evident).

Lines 393–394: This sentence should be rephrased or deleted.

Line 399: Please delete "(veterinary)".

In the discussion section, authors are expected to discuss their own data and compare them with those found in the literature. This section is very long; the authors should summarize it considerably. A few points that, in my opinion, could be deleted: 421-422; 487-498; 509-523.

Line 538: What do the authors mean by "advanced method"? Why would it be more advanced than those described in literature?

The level of English is good, the manuscript is easily readable and understandable.

Author Response

Answers to reviewer’s comments

Reviewer 3

Lines 18–20: This sentence is misleading. The authors could limit themselves to mentioning Q fever as a zoonosis.

Thank you for your statement. We rephrased the sentence, while also implementing suggestions from other reviewers as follows: “Inhalation of contaminated aerosols is the main route of transmission to humans in which C. burnetii can cause a persistent focalized infection with severe consequences.”

Line 21: The authors wrote "In a previous study, C. burnetii was detected in three dairy goat herds, but no control measures were implemented". This sentence is not necessary in the "simple summary" section.

We removed the phrase as suggested and slightly rephrased the following phrase to maintain understandability.

Line 27: "past infection". The presence of phase I antibodies could also indicate a chronic infection, not just a past infection.

There is still no decent definition for chronic C. burentii infection in ruminants. Therefore, we prefer to use «past infection». We changed this phrase while also including the findings in herd B more precisely: “In the other two herds, C. burnetii was shed to varying degrees and elevated antibody levels were present, indicating ongoing or past infection.”

Line 29: The authors wrote "And dust swabs from the milking parlor can serve as an easy diagnostic tool". I don't agree with this sentence because, although molecular biology applied to unconventional samples can give additional information in Q fever outbreaks, it is not possible to consider it a "diagnostic tool".

Thank you for addressing this point. We changed the sentence as follows: “In conclusion, the risk of C. burnetiishedding in dairy goat herds persists two years after the first detection, and dust swabs from the milking parlor can serve as an easy sampling tool.”

Line 32: "pathogen detection"? Maybe it would be better to convert it to outbreak."

Thank you for pointing this out to us. However, as stated in the discussion section, the results indicate that herd A did not experience a Q fever outbreak. Therefore, we suggest keeping the term “pathogen detection” to avoid disagreement with other statements in the study.

Line 43: The final sentence of the abstract could best summarize what was done and the impact of the new information obtained.

We revised the final sentence of the abstract in order to summarize the main points of the study as requested: “This study highlights the different infection dynamics in three unvaccinated dairy goat herds and the potential use of dust samples as a supportive tool to detect C. burnetii at herd level.”

Introduction:

Line 63: There is no reference to the exposure of the various ruminant species in Europe. In Europe, infection is present in cattle, small ruminants, buffaloes, and wild ruminants. I recommend this work to mention the presence of the infection also in the Mediterranean buffalo. doi: 10.3390/pathogens11080901

Thank you for pointing out the lack of this detail. We rephrased the sentence as follows: “Many species can transmit C. burnetii; however, in Europe, species such as cattle, goats, sheep, water buffaloes, and wild ruminants are the most important reservoirs.”

We added the reference concerning the presence of C. burnetii in water buffaloes as suggested.

Line 77: A reference to the Dutch outbreak, which, by the way, was mainly caused by goats, is needed. doi: 10.1016/j.medmal.2014.02.006

Thank you for your suggestion, we added the reference to the Dutch Q fever outbreak after the following sentence: “Spillover transmissions to humans are mostly linked to domestic small ruminants.” Also, we would like to point out that in this paragraph, another reference to the Dutch Q fever outbreak has already been used (Kampschreur et al. 2014, DOI: 10.1128/jcm.03221; as a reference to “Approximately 1-5% of cases develop a persistent focalized infection which is associated with endocarditis, vascular infection, and a possible fatal outcome.”

Line 104: I also recommend another reference for this sentence. doi: 10.1177/10406387221093581.

Thank you for your valuable comment. We added this reference after this sentence: “Commercial ELISA kits detect both, IgG phase I (PhI) and phase II (PhII) antibodies simultaneously.”

Some information in the Introduction should be streamlined and summarized.

We streamlined the introduction, however, our aim was to maintain as much information as possible to allow readers without profound knowledge of this research matter to follow the manuscript.

Material and methods:

Line 135: I’m not sure that "blended" is the appropriate term

There are multiple terms that refer to ELISA kits detecting IgG PhI and PhII antibodies simultaneously. The term “blended ELISA” has been previously introduced in literature to describe this type of ELISA (Bauer et al. 2021, DOI:10.1016/j.vaccine.2021.01.062 )

Line 137: "antigen". With the PCR described by the authors, they did not find the Coxiella antigen but its DNA.

We changed “antigen” to “C. burnetii DNA”.

Line 146: Do you have data on the productivity of these companies? Is it possible to carry out some kind of statistical analysis to highlight changes in productivity starting before the outbreak, during the outbreak, and in the following two years?

We agree that data such as productivity parameters are interesting aspects when investigating Q fever outbreaks. Unfortunately, a thorough analysis of productivity was beyond the scope of this study since its main focus was elsewhere. Also, the animal owners did not agree to share their production data with us or were not able to provide productivity data for this whole period. To end up, we also speculate that due to the COVID-19 pandemic, profitability as an alternative variable to describe productivity of the herd (e.g. money from selling milk) are hardly comparable to the years before the pandemic and therefore without greater value.

Line 318: I think the authors should delete "eartag loss".

We removed “eartag loss” as suggested.

Figure 1: This figure needs better definition or a better graphic design. A different graphic layout should also be taken into consideration for Figure 2 (for example, the cut-off line is not very evident).

Thank you for your comments about the readability of the graphics. We chose a black and white graphic design to keep the graphics accessible to individuals suffering from color blindness although it might hamper its readability.  We enhanced the cut-off line in figure 2 as suggested. We tried several options to enhance the readability of figure 1, especially to improve the threshold line visibility. Unfortunately, all our attempts resulted in a graphic that was harder to read. Thus, in accordance with the two other reviewers that did not wish for any changes in the graphic design of figure 1, we suggest to not alter it. Nevertheless, we added more details to the description of figure 1 to improve its understandability.

Lines 393–394: This sentence should be rephrased or deleted.

Thank you for pointing out this phrase. We deleted this phrase and rephrased the first part of the discussion as follows: “The present field study provided new data on the infection dynamics of C. burnetii in dairy goat herds where no vaccination program was implemented as a control measure. Such information is crucial for risk assessments for public health authorities and complements findings from herds with vaccination programs.”

Line 399: Please delete "(veterinary)".

We deleted “(veterinary)” as suggested.

In the discussion section, authors are expected to discuss their own data and compare them with those found in the literature. This section is very long; the authors should summarize it considerably. A few points that, in my opinion, could be deleted: 421-422; 487-498; 509-523.

Thank you for your valuable comments in order to streamline the discussion and shortened it according to your recommendations in accordance with the feedback from the two other reviewers. We deleted the sentence in line 421-422;

We significantly shortened lines 487-498 as follows: However, evaluating the bacterial viability in environmental specimens is challenging, still requires inoculation into live animals and is a time- and labor-intensive process.

Moreover, lines 509-423 were considerably shortened: “Overall, there was a strong correlation between vaginal excretion and milk shedding in herds A und C. However, in herd B, there was one negative BTM sample despite the presence of several goats shedding the pathogen through vaginal mucus. This discrepancy could be attributed to intermittent milk excretion. This highlights the necessity for repeated sampling to avoid false negative samples. In the present study, Chl. abortus was detected in abortion material in two goat herds (A, B). Additionally, a co-infection was detected with C. burnetii in herd B. Co-infections with both pathogens appear to occur regularly. Therefore, investigation for multiple abortifacient agents remains crucial even after the detection of C. burnetii.”

Line 538: What do the authors mean by "advanced method"? Why would it be more advanced than those described in literature?

Thank you for pointing this out to us. We deleted “using advanced methods”.

Comments on the Quality of English Language: The level of English is good, the manuscript is easily readable and understandable.

Round 2

Reviewer 3 Report

Thanks to the authors for addressing my comments. I believe that this stage of editing has greatly improved the quality of the manuscript. The work is almost ready to be considered for publication. Below are some of my minor comments.

Line 24: It would be more correct to write "measured in sera using phase-specific ELISA."

Figure 1 has no caption or description.

The text of some figures is really small. Authors may consider improving the overall quality of the images.

The English needs moderate editing in order to make the manuscript smoother and easier to understand.

Author Response

Reviewer 3 

Thanks to the authors for addressing my comments. I believe that this stage of editing has greatly improved the quality of the manuscript. The work is almost ready to be considered for publication. 

Below are some of my minor comments. 

Thank you for your additional input and we are sure the manuscript has been refined thanks to your comments. 

Line 24: It would be more correct to write "measured in sera using phase-specific ELISA." 

Thank you for your suggestion. We rewrote the phrase as suggested: “Antibody responses were measured in sera using phase-specific ELISAs. 

Figure 1 has no caption or description. 

Thank you for pointing this out. We rejoined the caption with the figure. 

The text of some figures is really small. Authors may consider improving the overall quality of the images. 

We appreciate your input concerning the figures. We used a bigger font size wherever possible and maximized the size of the figures. 

Comments on the Quality of English Language: The English needs moderate editing in order to make the manuscript smoother and easier to understand. 

Thank you for your comment. The manuscript underwent a thorough revision from a native speaker. 
